# Chemical Fingerprinting of Heat Stress Responses in the Leaves of Common Wheat by Fourier Transform Infrared Spectroscopy

**DOI:** 10.3390/ijms23052842

**Published:** 2022-03-04

**Authors:** Salma O. M. Osman, Abu Sefyan I. Saad, Shota Tadano, Yoshiki Takeda, Takafumi Konaka, Yuji Yamasaki, Izzat S. A. Tahir, Hisashi Tsujimoto, Kinya Akashi

**Affiliations:** 1United Graduate School of Agricultural Sciences, Tottori University, 4-101 Koyama-Cho-Minami, Tottori 680-0945, Japan; salmamohamedkhair@gmail.com (S.O.M.O.); shota.tdn@gmail.com (S.T.); takafumi.konaka@gmail.com (T.K.); 2Agricultural Research Corporation, Wad Medani P.O. Box 126, Sudan; sefian_ib@yahoo.com (A.S.I.S.); izzatahir@yahoo.com (I.S.A.T.); 3Faculty of Agriculture, Tottori University, 4-101 Koyama-Chou-Minami, Tottori 680-0945, Japan; b18a5097c@edu.tottori-u.ac.jp; 4Arid Land Research Center, Tottori University, 1390 Hamasaka, Tottori 680-0001, Japan; yujyamas@alrc.tottori-u.ac.jp (Y.Y.); tsujim@tottori-u.ac.jp (H.T.)

**Keywords:** wheat, heat stress, FTIR spectroscopy, chemometrics, arid region, linear discriminant analysis, spectral biomarker

## Abstract

Wheat (*Triticum aestivum* L.) is known to be negatively affected by heat stress, and its production is threatened by global warming, particularly in arid regions. Thus, efforts to better understand the molecular responses of wheat to heat stress are required. In the present study, Fourier transform infrared (FTIR) spectroscopy, coupled with chemometrics, was applied to develop a protocol that monitors chemical changes in common wheat under heat stress. Wheat plants at the three-leaf stage were subjected to heat stress at a 42 °C daily maximum temperature for 3 days, and this led to delayed growth in comparison to that of the control. Measurement of FTIR spectra and their principal component analysis showed partially overlapping features between heat-stressed and control leaves. In contrast, supervised machine learning through linear discriminant analysis (LDA) of the spectra demonstrated clear discrimination of heat-stressed leaves from the controls. Analysis of LDA loading suggested that several wavenumbers in the fingerprinting region (400–1800 cm^−1^) contributed significantly to their discrimination. Novel spectrum-based biomarkers were developed using these discriminative wavenumbers that enabled the successful diagnosis of heat-stressed leaves. Overall, these observations demonstrate the versatility of FTIR-based chemical fingerprints for use in heat-stress profiling in wheat.

## 1. Introduction

Wheat (*Triticum aestivum* L.) is one of the most important crops globally. Together with rice, maize, and soybean, these crops supply two-thirds of the calories that are required for the world population [1]. Wheat yield is sensitive to heat stress, and an approximately 6% loss in global yield is estimated with each Celsius degree increase in temperature caused by future global warming. The negative effects of heat stress on wheat yield are dependent upon the growth stages [2,3], and even a short duration of heat for one day reduced wheat yield [4]. Therefore, the development of new climate change adaptation measures that include the optimization of wheat cultivation practices and breeding of heat-tolerant wheat varieties is essential for the thermo-stressed regions of the globe [5]. Understanding the physiological and morphological responses to heat stress is of pivotal importance for genetic and/or agronomic improvement in wheat.

Metabolome techniques have provided an important tool for understanding environmental stress tolerance mechanisms in plants [6,7,8], and these techniques provide deeper insights into the phenotypic/agronomic variations influenced by the environment. Metabolome-based chemical fingerprinting has been employed as a selection tool for desirable traits in crops [7]. Various analytical platforms are available in the field of metabolomics, and multiple technologies are often required to gain comprehensive knowledge regarding the biochemical changes in each biological system [6]. Among the various metabolomic platforms, liquid chromatography-mass spectrometry (LC-MS) and gas chromatography-mass spectrometry (GC-MS) are the most widely used technologies to date. These methodologies have been successfully used for the characterization of complex metabolic responses in wheat, including changes under post-anthesis heat stress [8], growth-stage-specific metabolic responses to heat [9], and the effects of post-anthesis heat stress on the metabolic profile of the grain [10]. Although LC-MS- and GC-MS-based metabolomics exhibit the advantages of higher capacities for the detection and identification of metabolites, these techniques primarily target compounds possessing smaller molecular weights, and destructive extraction pretreatments are required prior to analyses.

In contrast, other metabolomic platforms, such as nuclear magnetic resonance (NMR) and Fourier transform infrared (FTIR) spectroscopy, possess the advantage of analyzing supramolecular structures such as cell walls with little sample preparation requirements [6,11,12]. FTIR spectroscopy possesses further advantages for potential applicability to in vivo imaging of biological materials [13,14] and remote sensing [15]. The FTIR spectroscopic technique has been used to study metabolic responses of plants to various environmental stresses, such as the salinity response in the beauty leaf tree (*Calophyllum inophyllum*) [16], and differential metabolic behaviors of roots and leaves in a halophyte *Sesuvium portulacastrum* under salt stress [17]. The FTIR spectroscopic technique has also been applied to different aspects of wheat such as metabolite distributions in the leaves under nitrate-limiting conditions [18], the oxidative-stress response of wheat roots [19], and structural changes in gluten [20], as well as for phylogenetic research examining cultivated and wild wheat species [21]. However, to our knowledge, no previous study has applied this technique to study metabolomic changes under heat stress in wheat. Therefore, the objective of the current study was to establish a protocol for fingerprinting and developing chemical biomarkers that characterize the molecular responses of common wheat to heat stress.

## 2. Results

### 2.1. Growth and Physiological Response of Wheat to Heat Stress

The common wheat cultivar ‘Norin 61’ was grown until the three-leaf stage at a daily temperature of 22 °C and then exposed to heat stress at a daily temperature of 42 °C for three days. A significantly higher canopy temperature of 37.1 °C ± 1.8 was observed in heat-stressed plants (hereafter designated as H3 plants) in comparison to 23.5 ± 1.9 °C in unstressed control plants of the same age (hereafter referred as C3 plants) and to 21.5 ± 2.0 °C in plants prior to the heat treatment (hereafter designated as C0 plants) (Figure 1A). The relative water content of the leaves was comparable between C3 and H3 plants (81.3 ± 9.8% and 77.4 ± 7.6%, respectively), and these values were not significantly different compared to that of C0 plants (85.9 ± 2.5%) (Figure 1B). Although the total leaf length increased from 72.8 ± 6.6 cm in the C0 plants to 88.2 ± 9.4 cm in H3 plants, it was significantly lower than that in C3 plants (102.0 ± 3.2 cm) (Figure 1C). Shoot biomass exhibited a similar trend, where the value for H3 plants (0.120 ± 0.019 g) was reduced by 17.7% in comparison to that in C3 plants (0.146 ± 0.009 g) (Figure 1D).

### 2.2. FTIR and Principal Component Analysis

The fully expanded third leaves of C3 and H3 plants were powdered, solidified as potassium bromide (KBr) pellets, and analyzed using FTIR spectroscopic technique. Figure 2 presents a typical example of the FTIR spectrum of each plant. These spectra exhibited largely similar patterns with a characteristic broad peak in the range of 2700–3700 cm^−1^, a number of sharper peak signals at approximately 2900 cm^−1^, complex contours in the 900–1800 cm^−1^ range, and relatively minor peak signals at approximately 400–800 cm^−1^ (Figure 2). The broad peak at approximately 2700–3700 cm^−1^ can be attributed to O–H, C–H, and N–H stretching, while the sharper peak signals at approximately 2900 cm^−1^ can be attributed to C–H stretching bands from aliphatic compounds [22]. In the so-called “finger-printing” region ranging from 400–1800 cm^−1^ [22], multiple peak signals are recognizable that largely overlapped and formed complex patterns. At least 12 peaks were detected in the spectra from both C3 and H3 plants, which can be assigned to various functional groups as shown in Table 1. However, it is noteworthy that the sample preparation conditions employed in this study, such as drying the leaf tissues at 70 °C, grinding, and the usage of a KBr matrix, might influence the wavenumber positions of maxima of some polar functional groups of biomolecules. Previous studies have demonstrated that the employment of a KBr matrix and grinding resulted in the shifts of some FTIR vibrational bands by up to 15 cm^−1^, which might influence the band energies, affect ion exchange, and induce crystallization of metastable amorphous biopolymers [23,24]. Nevertheless, from visual inspection, it was difficult to identify distinguishable features between C3 and H3 plants, and this suggested that the use of chemometric techniques is required for spectral analysis.

Subsequently, a principal component analysis (PCA) was employed to characterize the spectral differences between C3 and H3 plants. Figure 3A provides the PCA score plot of 358 spectra (180 and 178 spectra from C3 and H3 plants, respectively) that was based upon the variables of 3601 data points (normalized absorbance values from 400 to 4000 cm^−1^ with an interval of 1 cm^−1^) for each spectrum. The PC1-PC2 space in the plot explained 81.1% of the total variance (Figure 3A and Appendix A). Consequently, spectra from C3 and H3 plants were mostly clustered on the PC2-positive and negative half-planes, respectively, suggesting the presence of distinct spectral features between C3 and H3. Loading plots of the PCA showed complex patterns (Figure 3B–D); regions for PC2 loading over 0.5 were observed in wavenumbers of 459–484, 564–607, 610–614, 622–665, 670–752, 1177–1344, and 1351–1471 cm^−1^, whereas PC2 loading below −0.5 were seen in the regions of 2736–2897 and 2977–3082 cm^−1^ (Figure 3D), suggesting that absorbance of these specific positive and negative regions tended to influence separation of C3 and H3 plants. However, considerable numbers of C3 and H3 spectra were unseparated in the central origin of the score plot (Figure 3A), suggesting that the PCA alone was not sufficient to distinguish the spectral features in heat-stressed wheat leaves.

### 2.3. Linear Discriminant Analysis

A linear discriminant analysis (LDA) was employed to improve the discrimination of heat-stressed leaves. The 358 FTIR spectra that consisted of 180 and 178 spectra from C3 and H3 leaves, respectively, were randomly split into two groups at a ratio of 60:40%. The 60% group was used as a training set in the supervised machine learning process to construct a linear discriminant model. The LDA algorithm successfully separated the training dataset into the heat-stressed leaves from the controls in the histogram (Figure 4A), where the FTIR spectra with positive and negative LD1 scores corresponded to those taken from H3 and C3 leaves, respectively. The remaining 40% of the test dataset was then applied to the model for validation, and the results exhibited a slightly broader frequency distribution for both C3 and H3 in the histogram compared to those in the training set, while essentially confirming a clear discrimination between heat-stressed and control leaves (Figure 4B). Therefore, the FTIR spectral fingerprint in combination with the LDA approach was demonstrated to be effective in detecting discriminatory biochemical information in heat-stressed wheat leaves.

To assess which parts of the spectra were important for discriminating between heat-stressed and control leaves in LDA, the LDA loadings were examined. A plot of LDA loadings versus wavenumber revealed that several spectral regions, under a threshold of absolute loading intensity over 0.15, played key roles in regard to the discrimination ability (Figure 5). The plot exhibits two strong positive loading peaks at 1465 cm^−1^ (loading intensity of 0.398) and 1729 cm^−1^ (0.176) that contributed to the higher LDA score in the H3 leaves, and four strongly negative loading minimum points of 1251 cm^−1^ (loading intensity of −0.318), 576 cm^−1^ (−0.250), 1502 cm^−1^ (−0.224), and 482 cm^−1^ (−0.183) that contributed to the lower LDA score in the C3 leaves. These six spectral points were located within the multiple peak-overlapping region at 400–1800 cm^−1^ in the FTIR spectra (Figure 2) and corresponded to the finger-printing region [22]. These spectral regions may reflect changes in the chemical compositions and/or structures under heat stress that can potentially serve as spectral biomarkers for diagnosing heat-stress exposure in wheat leaves.

### 2.4. Spectral Biomarkers for Heat Stress Response

To explore the possibility of developing spectral biomarkers specific for the heat-stress response, the spectral regions that were identified as the major discriminants in the LDA loading plot presented in Figure 5 were further evaluated. For this purpose, two anchor points that encompass the target wavenumber were set, and a new parameter “Fm” (FTIR marker) that functions as a normalized target absorbance indicator was defined according to the offset absorbance values of the first and second anchors as 0 and 1, respectively (described in the Materials and Methods section below). The two anchor points were scanned in the vicinity of the target wavenumber and selected according to the following criteria: (i) the distance between the anchor point and target was within 150 cm^−1^; (ii) statistical significance (*p* value) of difference by Student’s *t*-test for Fm values between heat stress and control is below 0.0001; (iii) anchor points are preferably situated at visually discernible landmarks such as spectral peaks and minimum or inflection points within the spectral curves. The Fm values for the target wavenumber were calculated using 358 FTIR spectra data from C3 (180 spectra) and H3 (178 spectra) plants. Accordingly, anchor-1 and -2 were selected as shown in Table 2. A comparison of the averaged FTIR spectra between C3 and H3 plants in the magnified views showed that the target wavenumbers were mostly located in the middle of spectral slopes (Figure 6). The normalized absorbance at the target wavenumbers were slightly, but consistently, higher in Fm1465 and Fm1729 (Figure 6A,B), and lower in Fm1251, Fm576, Fm1502, and Fm482 (Figure 6C–F). Although knurl-like noises were detected in the FTIR spectra in the wavenumber range around 405–480 cm^−1^, a difference of absorbance at the target wavenumber of 482 cm^−1^ was notably larger than the fluctuation of the noises (Figure 6F). Box plots demonstrated that the biomarkers Fm1465 and Fm1729 exhibited significantly higher Fm values in H3 plants compared to those in C3 plants (Figure 7A, B, Table 2), while the other biomarkers (Fm1251, Fm576, Fm1502, and Fm482) possessed statistically lower Fm values in H3 plants compared to those in C3 plants (Figure 7C–F, Table 2). This was consistent with the positive and negative LDA loading values (Figure 5), respectively.

## 3. Discussion

### 3.1. Sensitivity of FTIR Spectral Response in Heat-Stressed Wheat Leaves

In this study, the FTIR spectroscopic technique was successfully applied to discriminate heat-stressed wheat leaves from those of control plants, thus demonstrating that this technique can serve as an analytical tool for monitoring chemical changes during heat stress in wheat. The heat stress applied in this study led to delayed leaf growth and biomass production (Figure 1C,D), and this was similar to observations reported in previous studies [27,28] where plants maintained their shoot growth to some degree under stress, thus suggesting that the intensity of the heat stress employed in this study was not at a lethal level. The relative water content in the leaves was statistically unchanged by heat stress in this study, unlike previously reported cases of heat-induced decline in wheat leaves [29,30]. This further indicated that the stress intensity in this study was relatively modest. Nevertheless, significant spectral differences were observed by FTIR spectroscopy, thus suggesting that FTIR-based fingerprinting was sensitive enough to characterize changes in the chemical constituents of wheat under non-lethal heat-stress conditions.

### 3.2. Chemometrics Using FTIR Spectra

As the FTIR spectra from heat-stressed and control plants were similar upon initial inspection (Figure 2), the application of chemometric methods was indispensable for obtaining a better interpretation of the FTIR spectra. Chemometric methods are commonly used to gain deeper insights from the obtained FTIR spectroscopic data [18,31,32,33]. As PCA alone was not sufficient to fully interpret the spectra (Figure 3), additional chemometric methods were applied. We applied LDA, which successfully discriminated between heat-stressed and control leaves, and we demonstrated the potency of the FTIR-based chemometric approach for diagnosing plant heat-stress status. Many previous studies have applied various chemometric methods. Johnson et al. (2003) [31] utilized PCA in combination with genetic algorithms to fingerprint salt-stressed tomato varieties. Recently, Nikalje et al. (2019) [17] applied PCA for characterizing metabolic responses of roots and leaves in a halophyte *S. portulacastrum*, demonstrating that FTIR spectroscopy differentiated different tissues and stress intensity in the PC1-PC2 plane. Cortizas and López-Costas (2020) [34] used PCA together with structural equation models to study the compositional and archaeological changes in human bone collagen. Grunert et al. (2020) [35] applied PCA for factor extraction, and this was followed by the use of two types of supervised machine learning methods (PCA-LDA and PCA-Mahalanobis discriminant analysis) for the classification of peritoneal dialysis effluent. Chemometric interpretation of FTIR spectra using the combination of PCA-LDA has also been utilized for the study of embryonic stem cell differentiation in murine models [36] and for the identification of spectral markers for putative stem cell regions of human intestinal crypts [37]. Consistent with these previous studies, the LDA applied in the present study successfully discriminated between heat-stressed and control leaves (Figure 4), thus demonstrating the potency of the FTIR-based chemometric approach for diagnosing plant heat-stress status.

### 3.3. FTIR-Based Biomarker for Chemical Changes under Heat Stress

The development of FTIR-based biomarkers has proven to be an effective analytical method in various scientific fields, including medical diagnosis [35], food quality control [32,33], and forensic analysis of cosmetic compounds [38]. In the present study, we developed FTIR-based spectral biomarkers that were based on the LDA loading intensity at specific wavenumbers (Figure 5). The developed biomarkers successfully distinguished heat-stressed leaves from controls (Table 1; Figure 7). Among the six biomarkers developed, Fm1465 and Fm1729 exhibited an increase under heat stress, while Fm1251, Fm576, Fm1502, and Fm482 exhibited a decrease under stress.

Among the biomarkers that increased under heat stress, the wavenumber for the marker Fm1465 was situated in the major region reported as a broad and poorly resolved C–H bending and C–O stretching region [22] that has been reported as a region for suberin/cutin in plant extracellular space [39,40]. The peak at 1465 cm^−1^ is also located in the vicinity of the reported assigned signals of 1463 cm^−1^ for CH_2_ scissoring and 1460 cm^−1^ for CH_3_ asymmetric bending in lipids [41] and the C–H signal in cell wall polysaccharides [22,42]. Responses of these candidate compounds in heat-stressed plants have been previously reported, including the complex regulation of leaf lipid composition in wheat [43] and heat stress-induced alteration of cell-wall components in the leaves of coffee [44] and wheat [45].

Another biomarker which increased under heat stress in this study was Fm1729. This wavenumber region can be assigned to stretching vibrations of ester C = O groups, which (together with the aforementioned bending C–H vibrations) are typical for lipids [24,25,26]. Similar increases in peak intensity around this region were observed in pea pollen grains under heat stress [46], which may indicate quantitative/qualitative regulation of the pollen exine layer under the stress. The increase in the Fm1729 value in wheat leaves in the present study may, therefore, suggest the adaptive alteration of lipid composition under heat stress. Alternatively, the increase in the Fm1729 value may indicate heat-induced injury in leaf lipids. Malondialdehyde (MDA), a major product of lipid peroxidation as a consequence of oxidative stress, has a characteristic FTIR signal around 1700–1750 cm^−1^ [47]. An increase of MDA was documented in wheat seedlings exposed to heat stress [48].

Among down-regulated Fm markers, the wavenumber of the Fm1251 marker was situated in the vicinity of the previously assigned signals of 1240 cm^−1^ for hemicellulose and 1260 cm^−1^ for pectin [25,49]. Pectin substances in the extracellular matrix have been demonstrated to function as a major regulatory factor for cell wall porosity in soybean cells [50], thus raising the hypothesis that adaptation to the heat environment may involve chemical rearrangement of pectins and foliar heat conductivity [44]. Other Fm markers that decreased under heat stress included Fm1502. A previous study by Kurian et al. [51] interpreted the wavenumber regions 1502–1600 cm^−1^ as aromatic skeletal vibration of lignin. Lima et al. [44] detected an alteration of lignin monomer composition after three days of heat stress in coffee leaves, suggesting that plant responses to the heat environment may include the structural rearrangement involving lignocellulose supramolecular structure.

Nevertheless, assignments of the proposed Fm biomarkers to any specific compounds are currently premature due to the intrinsic nature of overlapping signals in FTIR spectra and cumulative steric and/or electronic effects in a given molecule that can potentially lead to a large shift in spectral signals [22]. To identify the molecular entities for these Fm biomarkers, future biochemical and/or genetic studies are anticipated that may combine multifaceted approaches, including biomass fractionation, mass spectrometry, and genetic mapping.

### 3.4. Application of FTIR-Based Metabolome Profiling on Agronomy

The present study suggests that FTIR-based chemical fingerprinting can serve as a versatile tool for diagnosing plant physiological status under various environmental conditions, including heat stress. Metabolomics has been used as a powerful analytical tool to understand the links between agronomic performance and the underlying molecular mechanisms [6,7,8,52]. The versatility of FTIR spectroscopy has been demonstrated in previous studies in regard to discriminating genotype differences in cultivated and wild wheat species [21] and rice varieties [32]. Additionally, FTIR spectroscopy enables high-throughput measurements [53,54], suggesting that it can be used for chemo-typing heat-stress responses in crop breeding programs. Easier setup of FTIR spectroscopic facilities in comparison to that of other metabolomic platforms may also be beneficial for applying this technology to field metabolome studies [55,56]. Recently, FTIR-based remote sensing technologies have emerged as a new tool for monitoring the surface properties of land [15,57], and this may further broaden the possibility of developing spectrum-based plant diagnoses for crop production and breeding.

## 4. Materials and Methods

### 4.1. Plant Materials and Growth Conditions

Non-sterilized seeds from the common wheat cultivar ‘Norin 61’ were placed on a filter paper (qualitative filter paper No. 2, Advantec, Tokyo, Japan) that was cut out at an approximately 85-mm diameter in a Petri dish (88-mm diameter), and the seeds were imbibed by applying 6 mL of tap water to the dish so that the paper became evenly wet. Twelve seeds were placed per dish and the seed/water mass ratio was 1:15. The dish was covered by a transparent lid to avoid water evaporation, and the seeds were imbibed for three days at room temperature (25 °C) under a fluorescent room lamp illumination (a light intensity of approximately 10 µmol m^−2^ s^−1^) from 9 a.m. to 5 p.m. The germinated seeds were then transferred to pots (a height of 10 cm and diameter of 5 cm) containing 120 g of commercial horticulture soil (a brand “Oishii Yasaiwo Sodateru Baiyoudo”, Cainz, Honjo, Saitama, Japan) composed of composted bark, granular clay-like mineral, pumice, peat moss, perlite, and vermiculite. The soil was sterilized by autoclaving at 121 °C for 30 min before planting. Pots were transferred to a growth chamber with a 14/10 h day/night regime, a relative humidity setting of 50%, a light intensity of approximately 500 µmol m^−2^ s^−1^, and a temperature setting of 22/18 °C. Soil moisture level was maintained at 80–90% of field capacity (FC) [58] throughout the experiment. The 100% FC was determined as described previously [59]. When the plants reached the three-leaf stage and the length of the third leaf exceeded that of the second leaf, the seedlings were transferred to a heat chamber with a daily maximum temperature of 42 °C. In this heat chamber, the night temperature was 18 °C for 10 h, and the temperature setting was increased stepwise by 5 °C per hour from the beginning of the light regime for 3 h to a maximum temperature of 42 °C that was continued for 6 h. The temperature was then dropped to 33 °C for 1 h and then decreased stepwise by 5 °C per h to 18 °C during the next 3 h.

### 4.2. Measurements of Plant Growth and Physiology

Physiological measurements were acquired at three different conditions, including initial measurements on the day that treatment commenced (C0) and measurements at three days after the treatment for control (C3) and heat-stress (H3) conditions. Canopy temperature was measured using an FLIR-C2 thermal camera (FLIR system, Tallinn, Estonia). FLIR Tools software (v6.4.18039.1003) was used to estimate leaf surface temperature at 5 h after the beginning of the light regime.

For leaf relative water content measurement, the third leaf was harvested at 5 h after the beginning of the light regime, and a 2 cm leaf segment was obtained from the middle of the leaves. The fresh weight of the leaf segment was immediately measured using an electric balance, and turgid weight was measured after soaking the leaf segments in distilled water for 24 h at room temperature (25 °C). Tissue paper was used to remove the water from leaf surfaces before the turgid weight measurement. The leaf segments were transferred to an oven (EI-450B, ETTAS, AS-ONE, Osaka, Japan) at 70 °C to achieve complete dryness, and the dry weight was measured. Relative water content was calculated using the following formula [60];
100 × [(Fw − Dw)/(Tw − Dw)]
where Fw, Dw, and Tw denote the fresh weight, dry weight, and turgid weight of the leaf segment, respectively.

For the measurement of total leaf length, all leaves were harvested from the plants and scanned using a scanner (type DCP-J572N, Brother Industries, Nagoya, Japan). Leaf length was measured using ImageJ version 1.80 [61].

For biomass measurement, all aboveground tissues of the individual plants were dried in an oven at 70 °C until complete dryness. Their weights were measured after the samples were completely dried.

### 4.3. FTIR Measurement

The third leaf was harvested from the control and heat-treated plants and individually dried in an oven at 70 °C to achieve complete dryness. The dried leaves were ground into a fine powder using an agate mortar and pestle. The powdered samples (approximately 10 mg) were mixed with powdered KBr (IR grade, Nakalai, Kyoto, Japan) at a gravimetric ratio of 1:100, and approximately 10 mg of the mixture was transferred into a dice of 7 mm diameter in a hydraulic press (Pixie Hydraulic Pellet Press, PIKE Technologies, Madison, WI, USA). A pressure of 2.5 t cm^−2^ was then applied to form a thin disk. Ten disks were created from a single plant. FTIR absorbance spectra were recorded using a PerkinElmer Spectrum 65 spectrometer (Waltham, MA, USA) equipped with spectrum software version 10.4.2. Spectrum data were collected over the mid-infrared wavenumber range from 4000 to 400 cm^−1^ with a resolution of 1 cm^−1^ and 16 scans per measurement. Spectral measurements were repeated three times per disk, with an exception of one disk from the heat-stressed sample in which the measurement was performed only once. Data were collected from 60 disks where each was derived from six plants each from control and heat-stressed plants, and 180 and 178 spectral data were obtained for control and heat-stressed leaves, respectively.

### 4.4. Chemometrics of Spectral Data

FTIR spectra were baseline-corrected using a linear gradient of absorbance values at 4000 and 400 cm^−1^, and the absorbance values were normalized to obtain a total value of 1 million for each spectrum. A principal component analysis (PCA) was performed using the prcomp function in the stat package (v3.6.2) in R Statistical Software [62], and the score plot and loading plot was drawn using the ggplot function in the ggplot2 package (version 3.3.5) in R. For LDA, the 358 spectral datasets in the range from 3600 to 400 cm^−1^ wavenumber were randomly split into a training and test set at a ratio of 60% to 40% using the sample function in the base package (v3.6.2) in R and then calculated using the lda function in the MASS package (v7.3–54).

For the development of spectral Fm biomarkers, a custom-made R script was written to scan the two candidate anchor point wavelengths in the 300 cm^−1^ range spanning the target wavenumber and for calculating the Fm values and *p*-value in the Student’s *t*-test. The Fm values were calculated using the following formula:Fm = (*A_target_* − *A*_*anchor*1_)/(*A*_*anchor*2_ − *A*_*anchor*1_)
where *A_target_*, *A_anchor_*_1_, and *A_anchor_*_2_ denote the normalized absorbance values for the target and *anchors* 1 and 2, respectively. The R scripts were deposited in Appendix A.

### 4.5. Statistical Analysis

The *t*-test function in the stats package (v3.6.2) was used for Student’s *t*-test. Tukey’s test was performed using the Astatsa.com online web statistical calculator [63].

## 5. Conclusions

In the present study, an FTIR-based fingerprint technique was applied to characterize the metabolome response of wheat leaves to heat stress. Although PCA was unable to achieve complete separation of stressed leaves from controls, LDA clearly discriminated between these two samples, thus demonstrating that LDA-based chemometrics using FTIR spectra provides a powerful approach for monitoring heat-induced chemical changes in wheat leaves. Based on the markedly altered spectral fingerprinting regions, six spectral biomarkers were developed that correctly reflected the heat-stress status of the leaves. Overall, the present study suggests the potential of FTIR spectroscopy, coupled with chemometrics analysis, for studying the heat-stress response and tolerance mechanisms in wheat.

## Figures and Tables

**Figure 1 ijms-23-02842-f001:**
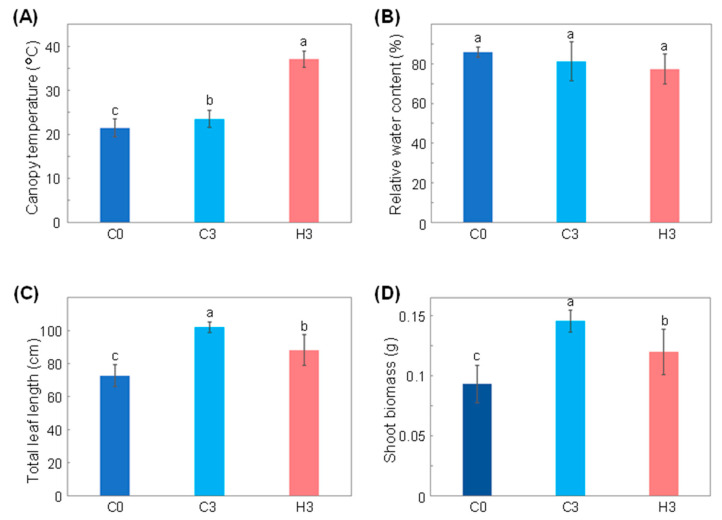
Impact of heat stress on wheat growth and physiology. (**A**) The canopy temperature, (**B**) relative water content, (**C**) total leaf length, and (**D**) shoot biomass of plants prior to heat treatment (C0), of control plants after three days (C3), and of plants subjected to heat stress for three days (H3) are presented. Values are the average and standard deviation for 18–31 measurements from 5–6 plants in “(**A**)” and for 5–6 plants in “(**B**–**D**)”. Statistical analysis was carried out by Tukey’s range test (*p* < 0.05) and different letters (a, b, and c) were used to indicate significant differences between treatments.

**Figure 2 ijms-23-02842-f002:**
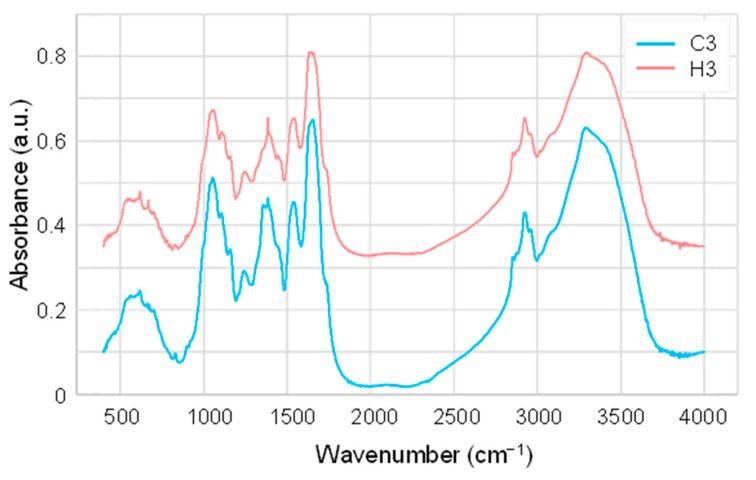
Representative FTIR spectra from the leaves of control (C3) and heat-stressed (H3) wheat.

**Figure 3 ijms-23-02842-f003:**
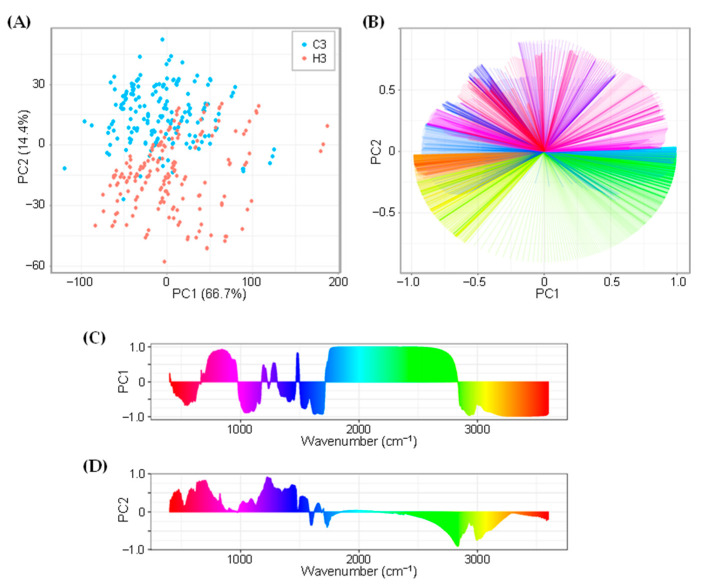
Principal component analysis of FTIR spectra. (**A**) A score plot showing overlapping distribution between C3 and H3 plants. (**B**) A two-dimensional loading plot. Assignment of a color gradient to respective wavenumbers are the same as those presented in (**C**,**D**). (**C**,**D**) One-dimensional loading column plots for (**C**) PC1 and (**D**) PC2. The loading for each wavenumber is expressed using a color gradient image along their *x*-axes.

**Figure 4 ijms-23-02842-f004:**
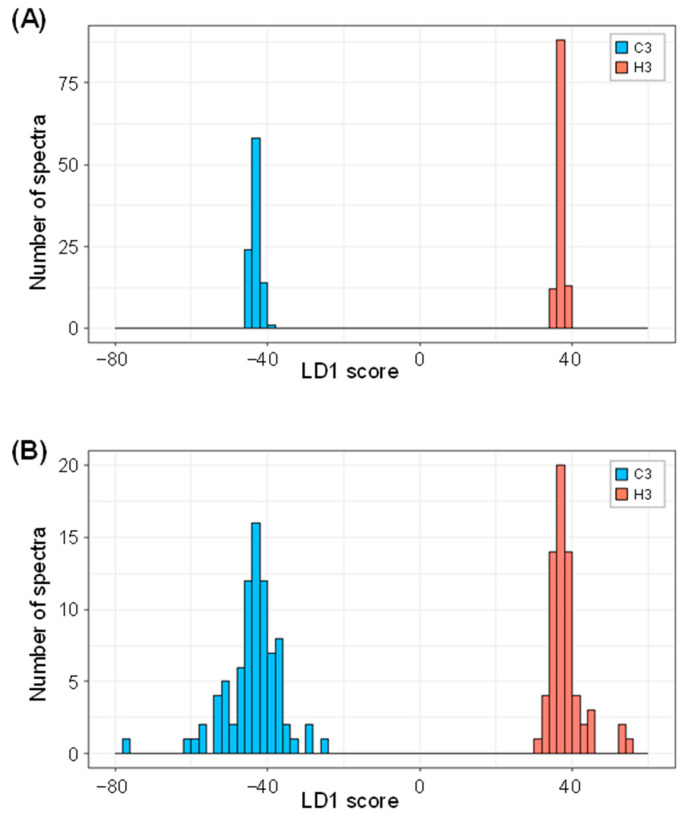
Two group histograms of the training (**A**) and the test (**B**) sets for FTIR spectra based on the LD1 score in the linear discriminant analysis, demonstrating classification performance between control and heat-stressed leaves.

**Figure 5 ijms-23-02842-f005:**
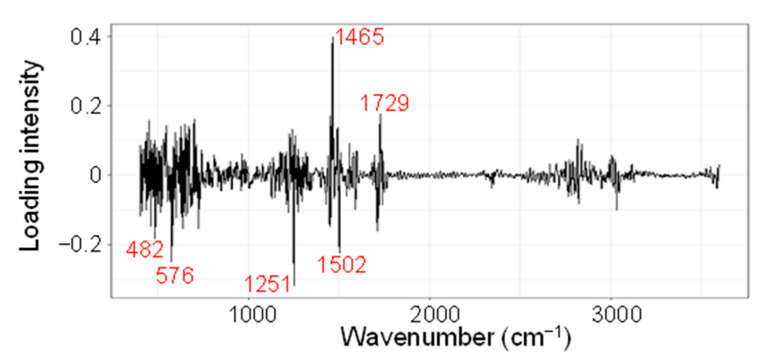
Identification of the discriminatory spectral region. Loading plot of the LDA results that were used for the classification of H3 and C3 leaves. Wavenumbers for the major peaks and minimum turning points are indicated by red fonts.

**Figure 6 ijms-23-02842-f006:**
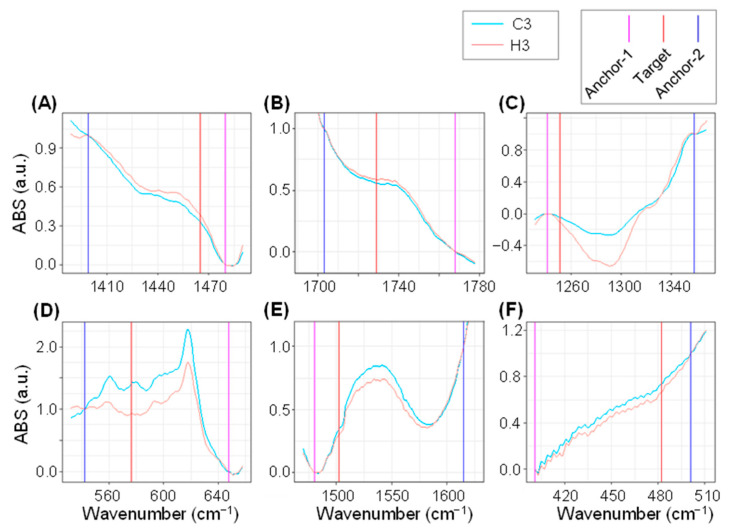
Magnified view of averaged FTIR spectra in the vicinity of Fm biomarkers. Normalized spectra for C3 (cyan) and H3 (orange) plants in the vicinity of (**A**) Fm1465, (**B**) Fm1729, (**C**) Fm1251, (**D**) Fm576, (**E**) Fm1502, and (**F**) Fm482 markers are shown. Red, magenta, and blue vertical lines designate the locations of wavenumbers for the target, anchor-1, and -2, respectively.

**Figure 7 ijms-23-02842-f007:**
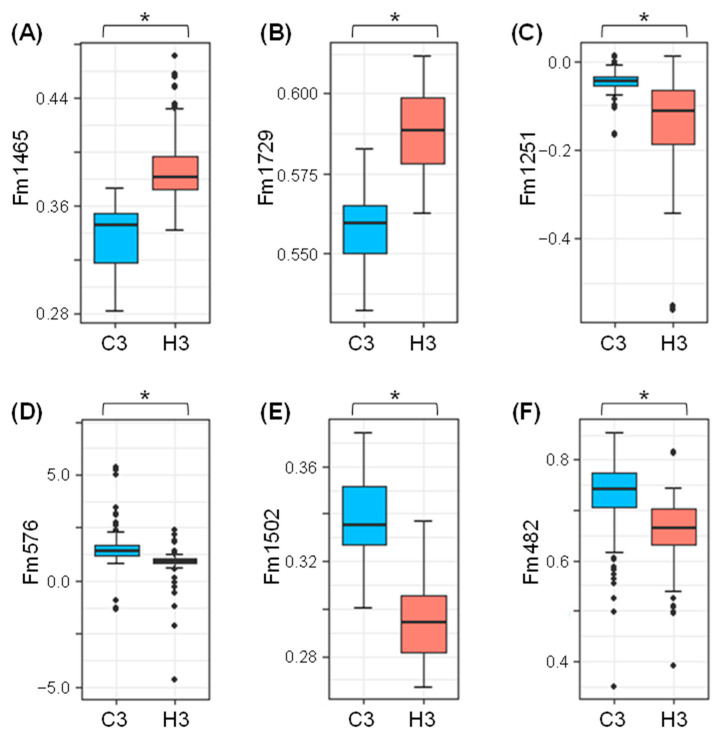
Box plots presenting the comparison of spectral biomarker values between C3 and H3 plants. Plots for the spectral biomarkers (**A**) Fm1465, (**B**) Fm1729, (**C**) Fm1251, (**D**) Fm576, (**E**) Fm1502, and (**F**) Fm482 are provided. The 180 and 178 spectra were used for the C3 and H3 plots, respectively. Asterisks represent statistically significant differences at *p* < 0.001.

**Table 1 ijms-23-02842-t001:** Major FTIR peaks observed and their assignment to probable functional groups in wheat leaves.

No.	Wavenumber (cm^−1^)	Probable Functional Groups
1	3293	O–H stretching, N–H stretching.
2	2960	C–H stretching in –CH_3_ (antisymmetric).
3	2925	C–H stretching in –CH_2_– (antisymmetric).
4	2852	C–H stretching in –CH_2_– (symmetric).
5	1651	C=C stretching, C=O stretching (amide), N–H bending (amide I).
6	1541	C=C stretching (aromatic), N–H bending (amide II), C–N stretching.
7	1385	C–H bending (antisymmetric), =C–H in-plain bending.
8	1241	C–O stretching, in-plain C–H bending (aromatic), aliphatic C–O stretching, P=O stretching (aliphatic).
9	1158	C–O stretching, C–N stretching (aliphatic), in-plain C–H bending (aromatic), aliphatic C–O stretching.
10	1106	C–O stretching, C–N stretching (aliphatic), in-plain C–H bending (aromatic), aliphatic C–O stretching.
11	1055	C–O stretching, C–N stretching (aliphatic), in-plain C–H bending (aromatic).
12	618	=C–H out-of-plane bending, =C–H bending, C–S stretching.

Assignment of wavenumbers to probable functional groups are according to [24,25,26].

**Table 2 ijms-23-02842-t002:** Characteristics of spectral biomarkers.

Marker Name	Wavenumbers (cm^−1^)			Loading *^1^	Median Fm Value		H3/C3 Ratio *^2^	*p ** ^3^
	Target	Anchor-1	Anchor-2		C3	H3		
Fm1465	1465	1480	1399	0.398	0.345	0.381	1.104	2.1 × 10^−61^
Fm1729	1729	1768	1703	0.176	0.559	0.588	1.052	3.8 × 10^−80^
Fm1251	1251	1241	1358	−0.318	−0.0428	−0.112	2.607	3.1 × 10^−26^
Fm576	576	648	542	−0.25	1.436	0.899	0.626	1.1 × 10^−4^
Fm1502	1502	1480	1615	−0.224	0.335	0.294	0.879	4.4 × 10^−75^
Fm482	482	401	501	−0.183	0.741	0.666	0.899	7.9 × 10^−21^

^*1^ Loading score of LDA at the target wavenumber. *^2^ H3/C3 ratio of median Fm values. *^3^ Probability by *t*-test.

## Data Availability

Not applicable.

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
