# Peer review of "Chemical Fingerprinting of Heat Stress Responses in the Leaves of Common Wheat by Fourier Transform Infrared Spectroscopy"

_ijms, 2022, doi:10.3390/ijms23052842_

Round 1

Reviewer 1 Report

In this work, the effect on an increase in ambient temperature on the three-leaf stage wheat plants was evaluated using mainly FTIR spectroscopy. The work is of great importance to the agriculture regarding to global warming. The authors do a good introduction on the subject and the importance of this study, although I believe that a further discussion on the latest works would be beneficial. For instance, this work https://doi.org/10.1016/j.btre.2019.e00352 appears to be related to their work and could also be used in the discussion.

In detail, in figure 1, it is not clear what the letter a, b and c over the columns represent. It would be important to explain in the text what they represent, and maybe use a different symbol, since it gets also confusing with the a, b and c that enumerate the panels of the figure.

It is really interesting the fact that already the PCA is showing a separation. It would be useful to have a look to the PC2 loading, which seems to be the significant one.

From the results of LDA, it is not clear if it was based on single wavelength, was any threshold applied? What is your opinion on the importance of figure 5 on the main text? While the peaks appearing over 1000 cm-1 seem quite clear, under this wavelength it appears noisy. I was wondering if the results would be reproducible after applying some smoothing algorithm. From these results, it derives what has been defined as marker name (Fm). I understand it as integration of significant bands. In this case, I think it would be useful to have these areas highlighted over an average spectrum in order to see how the edges of the integration are covering a significant band. This supports my doubt on the significance of the two contributions under 1000 cm-1 (576 cm-1 and 482 cm-1) because of the noise observed in that region and because those are the ones that show a smaller difference.

What could be a possible explanation of the decrease of Fm1502 under heat stress?

In general, I find very interesting the article, and I would support its publication after these questions are addressed.

Author Response

We greatly appreciate your careful consideration and helpful suggestions on our manuscript of the

Manuscript ID: ijms-1598793, entitled “Chemical Fingerprinting of Heat Stress Responses in the Leaves of Common Wheat by Fourier-Transform Infrared Spectroscopy”, which was submitted as Article to the International Journal of Molecular Sciences. Please find enclosed the revised version of our manuscript. The portions of the revisions made in the manuscript file are shown in red font.

In the section below, the comments given to the manuscript by the Reviewer are described in italics, and our responses to the comments are described point-by-point.

Comment on the manuscript:

  1. In this work, the effect on an increase in ambient temperature on the three-leaf stage wheat plants was evaluated using mainly FTIR spectroscopy. The work is of great importance to the agriculture regarding to global warming. The authors do a good introduction on the subject and the importance of this study, although I believe that a further discussion on the latest works would be beneficial. For instance, this work https://doi.org/10.1016/j.btre.2019.e00352 appears to be related to their work and could also be used in the discussion.

Response to the Comment:

We thank to the Reviewer for the helpful suggestions. According to the comment, the suggested work and other recent work were incorporated both in the introduction and discussion.

In the introduction, following sentence was added in lines 66-70:

“The FTIR method has been used to study metabolic responses of plants to various environmental stresses, such as the salinity response in beauty leaf tree (Calophyllum inophyllum) [16], and differential metabolic behaviors of roots and leaves in a halophyte Sesuvium portulacastrum under salt stress [17].”

Moreover, following sentence was incorporated in the discussion (lines 267-269):

“Recently, Nikalje et. al. (2019) [17] applied PCA for characterizing metabolic responses of roots and leaves in a halophyte S. portulacastrum, demonstrating that FTIR differentiated different tissues and stress intensity in the PC1-PC2 plain.”

Comment on the manuscript:

  1. In detail, in figure 1, it is not clear what the letter a, b and c over the columns represent. It would be important to explain in the text what they represent, and maybe use a different symbol, since it gets also confusing with the a, b and c that enumerate the panels of the figure.

Response to the Comment:

We apology our mistake in the figure style in the previous version of the manuscript, in which the panel signs and statistical signs has been mixed up with the same small letters of “a”, “b”, “c”. In the revised version, the panel signs were changed to the capital letter throughout the manuscript. Moreover, an explanation of the small letters (“a”, “b”, “c”) for statistical significance was included in lines 101-102 as below:

“Statistical analysis was carried out by Tukey’s range test (p < 0.05) and different letters (a, b, and c) were used to indicate significant differences between treatments.”

Comment on the manuscript:

  1. It is really interesting the fact that already the PCA is showing a separation. It would be useful to have a look to the PC2 loading, which seems to be the significant one.

Response to the Comment:

We thank the Reviewer for the interests on the PCA data. As suggested, the loading data was added to Figure 3. Since the 2D-loading plot in Figure 3B showed a very complex pattern, we extracted PC1 and PC2 loading data and presented as 1D-column plots in Figure 3C and D, and a list of peak maxima and minima for the loading plots in Supplementary Table 1. The data in Figure 3D suggested the contribution of specific wavenumber regions to the separation of C3 and H3 plants in PCA analysis. Accordingly, following descriptions were presented in lines 140-145 in the revised manuscript.

“Loading plots of the PCA showed the complex patterns (Figure 3B–D); regions for PC2 loading over 0.5 were observed in wavenumbers of 459–484, 564–607, 610–614, 622–665, 670–752, 1177–1344, and 1351–1471 cm-1, whereas PC2 loading below -0.5 were seen in the regions of 2736–2897 and 2977–3082 cm-1 (Figure 3D), suggesting that absorbance of these specific positive and negative regions tended to influence separation of C3 and H3 plants.”

Comment on the manuscript:

  1. From the results of LDA, it is not clear if it was based on single wavelength, was any threshold applied? What is your opinion on the importance of figure 5 on the main text? While the peaks appearing over 1000 cm-1 seem quite clear, under this wavelength it appears noisy. I was wondering if the results would be reproducible after applying some smoothing algorithm. From these results, it derives what has been defined as marker name (Fm). I understand it as integration of significant bands. In this case, I think it would be useful to have these areas highlighted over an average spectrum in order to see how the edges of the integration are covering a significant band. This supports my doubt on the significance of the two contributions under 1000 cm-1 (576 cm-1 and 482 cm-1) because of the noise observed in that region and because those are the ones that show a smaller difference.

Response to the Comment:

We appreciate the concerns from the Reviewer regarding the LDA. The LDA was performed using the 358 spectral dataset spanning from wavenumbers of 3,600 to 400 cm-1. This information is added to the revised manuscript (lines 417-418). We employed a threshold absolute value of 0.15 for the loading intensity in the LDA. This information is inserted in the revised manuscript (lines 178-179). We think that the loading plot of the LDA presented in Figure 5A is of pivotal importance to search for the location of discriminant factors between control and heat stressed leaves in the FTIR spectra. However, the Figure 5B in the previous manuscript, which showed a representative spectrum as a reference, was somewhat repetitive to that in Figure 2 so that we withdraw the Figure 5B from the revised manuscript.

                 As suggested by the Reviewer, we added new data in Figure 6 for the highlighted, magnified view of averaged spectra in the vicinity of 6 Fm markers in the revised manuscript. The 6 panels in the new Figure 6 all showed consistently the difference of absorbance between heat-stressed and control leaves at the target wavenumbers; the differences were apparently larger than the observed fluctuations caused by the noises, including the 576 cm-1 and 482 cm-1 that were noisy in the LDA loading plot. Therefore we reasoned that the latter two markers were significant. These observations were described in lines 212-219 of the revised manuscript as follow:

 “Comparison of averaged FTIR spectra between C3 and H3 plants in the magnified views showed that the targets wavenumbers were mostly located in the middle of spectral slopes (Figure 6). The normalized absorbance at the target wavenumbers were slightly but consistently higher in Fm1465 and Fm1729 (Figure 6A, B), and lower in Fm1251, Fm 576, Fm 1502, and Fm482. Although knurl-like noises were detected in the FTIR spectra in the wavenumber range around 405–480 cm-1, difference of absorbance at the target wavenumber of 482 cm-1 was notably larger than the fluctuation of the noises (Figure 6F).”

                 Smoothing algorism can be an option for decreasing noises in FTIR, but at the same time it decreases the amount of information in the spectra, as has been documented for plant leaf samples (https://doi.org/10.1016/j.chemolab.2011.07.001). We did not employ smoothing algorism, because in our preliminary experiments we observed significantly diminished signals for smaller FTIR peaks.

Comment on the manuscript:

  1. What could be a possible explanation of the decrease of Fm1502 under heat stress?

Response to the Comment:

One possible explanation for the behavior of Fm1502 could be on the physicochemical change of lignin, because the wavenumber 1502 cm-1 was previously assigned as aromatic skeletal vibration of lignin. Therefore, following descriptions were added in lines 317-322 of the revised manuscript:

“Other Fm markers which decreased under heat stress included Fm1502. Previous study by Kurian et. al. [43] interpreted the wavenumber regions 1502-1600 cm-1 as aromatic skeletal vibration of lignin. Lima et. al. [39] detected alteration of lignin monomer composition after three days of heat stress in coffee leaves, suggesting that plant responses to heat environment may include structural rearrangement involving lignocellulose supramolecular structure.”

Reviewer 2 Report

The manuscript presents an analysis of FTIR spectroscopic data for heat-stressed wheat leaves as compared with the control tissues, which show that even a moderate heat stress can be detected using LDA of the data. Such an application of FTIR spectroscopy to this important agricultural problem definitely deserves attention. The material is of interest for a relatively broad general readership, of good potential impact and importance within the broad field of biomolecular sciences. The manuscript is professionally written, clear and the results are relatively straightforward. However, several points should be clarified before the paper can become publishable, as listed below (the commented text is referred to by corresponding manu. line numbers in parentheses).

(17-18) “Fourier transform infrared (FT-IR)” – COMMENT: In the title (and in text, line 62), the authors use “Fourier-transform” (with a hyphen after “Fourier”), while in the abbreviation, the hyphen is put as “FT-IR” (?). Since in the majority of cases in the literature the term is spelled and abbreviated without a hyphen (see, e.g. your Refs. [11-13, 15, 17, 26, 27, 31-33, 36], etc.), it would be reasonable not to use a hyphen both in the term and in the abbreviation throughout the manu.: “Fourier transform infrared (FTIR)”.

(62-63) The phrase “biological polymers such as cell walls” is ‘awkward’ and in fact wrong: cell walls are supramolecular structures and are not biopolymers! The terminology should be accurate: even proteins are not biopolymers (as they have no single monomer whose polymerization would give a polymer, i.e. a macromolecule with repeating monomers throughout the molecular chain), they are biomacromolecules. Thus, it would be correct to say that cell walls include biomacromolecules. Rephrase in a correct way for clarity.

(96-106) In this part of subsec. 2.2, it is necessary to mention for clarity that the use of a KBr matrix (i.e., pressed pellets of a powdered dried sample mixed with KBr powder) and even grinding itself could influence the positions of maxima of some polar functional groups of biomolecules, as has been documented (see, e.g. https://doi.org/10.1016/j.saa.2017.12.051 and https://doi.org/10.3390/molecules26041146 ). It is also notable that the use of 70 degrees C for drying the leaves (see line 318 of subsec. 4.3) could affect the structure of leaf proteins (e.g. via partial thermal denaturing), which could be reflected in their typical amide I and amide II peaks (around ~1630-1660 and ~1530-1550 cm-1, respectively; see Fig. 2). For more informativity, in Fig. 2 it would be reasonable to indicate wavenumbers of peak maxima for all the main peaks (the same for Fig. 5,b).

(147-148) The “two strong positive loading peaks at 1,465 cm-1 … and 1,729 cm-1” in fact can be assigned to bending vibrations in methyl/methylene groups (as is discussed in lines 237-243) and to stretching vibrations in ester C=O groups which both are typical of lipids. Please add the band at 1729 cm-1 (stretching C=O in lipids) to the discussion in subsec. 3.3.

(278-282) “Seeds from the common wheat cultivar ‘Norin 61’ were imbibed in a Petri dish” – COMMENT: the description in this part of subsec. 4.1 is insufficient. What was the seeds-to-water ratio in the Petri dishes? Which water (or solution) was used (purity)? Were the seeds sterilised or not? Was the commercial horticulture soil sterilized? Which was its water content? Was it maintained during wheat growth? These data are important for the experiment to be reproducible (note that the microbial consortia of both seeds (on their surface) and in the soil might influence the growth and composition of the plant tissues).

(302) “turgid weight was measured after soaking the leaf segments in distilled water for 24 h” – COMMENT: It is not clear whether any possible surface water drops were somehow removed from the surface of the wet leaf segment, e.g. with a filter paper (right after soaking and prior to weighing). The “turgid weight” is probably supposed to include the imbibed water (inside the plant tissue), but not the surface water layer.

(SUPPL. INFO) I would recommend to combine all the SI in one PDF file (to facilitate its use).

Author Response

We greatly appreciate your careful consideration and helpful suggestions on our manuscript of the

Manuscript ID: ijms-1598793, entitled “Chemical Fingerprinting of Heat Stress Responses in the Leaves of Common Wheat by Fourier-Transform Infrared Spectroscopy”, which was submitted as Article to the International Journal of Molecular Sciences. Please find enclosed the revised version of our manuscript. The portions of the revisions made in the manuscript file are shown in red font.

In the section below, the comments given to the manuscript by the Reviewer are described in italics, and our responses to the comments are described point-by-point.

Comment on the manuscript:

  1. (17-18) “Fourier transform infrared (FT-IR)” – COMMENT: In the title (and in text, line 62), the authors use “Fourier-transform” (with a hyphen after “Fourier”), while in the abbreviation, the hyphen is put as “FT-IR” (?). Since in the majority of cases in the literature the term is spelled and abbreviated without a hyphen (see, e.g. your Refs. [11-13, 15, 17, 26, 27, 31-33, 36], etc.), it would be reasonable not to use a hyphen both in the term and in the abbreviation throughout the manu.: “Fourier transform infrared (FTIR)”.

Response to the Comment:

As suggested by the Reviewer, the hyphen was removed and the terminology “FTIR” was used throughout the revised manuscript.

Comment on the manuscript:

  1. (62-63) The phrase “biological polymers such as cell walls” is ‘awkward’ and in fact wrong: cell walls are supramolecular structures and are not biopolymers! The terminology should be accurate: even proteins are not biopolymers (as they have no single monomer whose polymerization would give a polymer, i.e. a macromolecule with repeating monomers throughout the molecular chain), they are biomacromolecules. Thus, it would be correct to say that cell walls include biomacromolecules. Rephrase in a correct way for clarity.

Response to the Comment:

We appreciate very much to the Reviewer for correcting our naive mistake. As suggested, we rephrase the portions as “analyzing supramolecular structures such as cell walls” in the revised manuscript.

Comment on the manuscript:

  1. (96-106) In this part of subsec. 2.2, it is necessary to mention for clarity that the use of a KBr matrix (i.e., pressed pellets of a powdered dried sample mixed with KBr powder) and even grinding itself could influence the positions of maxima of some polar functional groups of biomolecules, as has been documented (see, e.g. https://doi.org/10.1016/j.saa.2017.12.051 and https://doi.org/10.3390/molecules26041146 ). It is also notable that the use of 70 degrees C for drying the leaves (see line 318 of subsec. 4.3) could affect the structure of leaf proteins (e.g. via partial thermal denaturing), which could be reflected in their typical amide I and amide II peaks (around ~1630-1660 and ~1530-1550 cm-1, respectively; see Fig. 2). For more informativity, in Fig. 2 it would be reasonable to indicate wavenumbers of peak maxima for all the main peaks (the same for Fig. 5,b).

Response to the Comment:

We thank the reviewer very much for pointing out the important points. As suggested, the risks for employing KBr matrix, grinding, and oven dry procedures were described in lines 117-123 in the revised manuscript, together with the suggested two references. Moreover, the wavenumbers of detected peak maxima were listed in a new Table 1 in the revised manuscript (lines 115-117).

Comment on the manuscript:

  1. (147-148) The “two strong positive loading peaks at 1,465 cm-1 … and 1,729 cm-1” in fact can be assigned to bending vibrations in methyl/methylene groups (as is discussed in lines 237-243) and to stretching vibrations in ester C=O groups which both are typical of lipids. Please add the band at 1729 cm-1 (stretching C=O in lipids) to the discussion in subsec. 3.3.

Response to the Comment:

As suggested by the Reviewer, description on the wavenumber 1729 cm-1 was added in lines 301-311 as follow:

“Another biomarker which increased under heat stress in this study was Fm1729. This wavenumber region can be assigned to bending vibrations in methyl/methylene groups and to stretching vibrations in ester C=O groups, which both are typical for lipids [stuart]. Similar increases in peak intensity around this region was observed in pea pollen grains under heat stress [45], which may indicate quantitative/qualitative regulation of pollen exine layer under the stress. The increase in the Fm1729 value in wheat leaves in the present study may therefore suggest the adaptive alteration of lipid composition under heat stress. Alternatively, the increase in the Fm1729 value may indicate heat-induced injury in leaf lipids. Malondialdehyde (MDA), a major product of lipid peroxidation as a consequence of oxidative stress, has a characteristic FTIR signal around 1700-1750 cm-1 [46]. Increase of MDA was documented in wheat seedlings exposed to heat stress [47].”

Comment on the manuscript:

  1. (278-282) “Seeds from the common wheat cultivar ‘Norin 61’ were imbibed in a Petri dish” – COMMENT: the description in this part of subsec. 4.1 is insufficient. What was the seeds-to-water ratio in the Petri dishes? Which water (or solution) was used (purity)? Were the seeds sterilised or not? Was the commercial horticulture soil sterilized? Which was its water content? Was it maintained during wheat growth? These data are important for the experiment to be reproducible (note that the microbial consortia of both seeds (on their surface) and in the soil might influence the growth and composition of the plant tissues).

Response to the Comment:

We appreciate the Reviewer for pointing out the ambiguous description regarding wheat planting and cultivation. As suggested by the Reviewer, more explanation was added to lines 349-359 and 361-362 in the section 4.1. as follow:

“Non-sterilized Seeds from the common wheat cultivar ‘Norin 61’ were  placed on a filter paper (qualitative filter paper No. 2, Advantec, Tokyo, Japan) which was cut out at approximately 85-mm diameter in a Petri dish (88-mm diameter), and the seeds were imbibed by applying 6 ml of tap water to the dish so that the paper became evenly wet. Twelve seeds were placed per dish and the seed/water ratio was 1:15. The seeds were imbibed for three days at room temperature (25 °C), and then germinated seeds were transferred to pots (a height of 10 cm and diameter of 5 cm) containing 120 g of commercial horticulture soil (a brand “Oishii Yasaiwo Sodateru Baiyoudo,” Cainz, Honjo, Saitama, Japan) containing composted bark, granular clay-like mineral, pumice, peat moss, perlite, and vermiculite. The soil was sterilized by autoclaving at 121 °C for 30 minutes before planting.”

“Soil moisture level was maintained at 80–90% of field capacity throughout the experiment.”

Comment on the manuscript:

  1. (302) “turgid weight was measured after soaking the leaf segments in distilled water for 24 h” – COMMENT: It is not clear whether any possible surface water drops were somehow removed from the surface of the wet leaf segment, e.g. with a filter paper (right after soaking and prior to weighing). The “turgid weight” is probably supposed to include the imbibed water (inside the plant tissue), but not the surface water layer.

Response to the Comment:

We apology that an information was not described regarding how to remove excess water during turgid weight measurement in the previous version of the manuscript. We used tissue papers for removing the excess water. As suggested by the Reviewer, additional information was added to lines 380-381 in the section 4.2. as follow:

“Tissue paper was used to remove the water from leaf surfaces before the turgid weight measurement.

Comment on the manuscript:

  1. (SUPPL. INFO) I would recommend to combine all the SI in one PDF file (to facilitate its use).

Response to the Comment:

As suggested by Reviewer, supplementary data was combined in one PDF file.

Round 2

Reviewer 2 Report

The manuscript has been substantially improved during revision. Nevertheless, before it could become acceptable, a few further minor but important corrections (mainly in the newly added parts) are necessary (listed below; the line numbers (in parentheses) are cited from the revised version, i.e. version 2).

(Lines 63, etc.) The phrase “Fourier-transform infrared (FTIR)” is actually an adjective (with its abbreviation). The whole term you mean (and which should be used in the correct technical language) is “Fourier-transform infrared (FTIR) spectroscopy” (or further, in the abbreviated form, “FTIR spectroscopy” (commonly used without an article) or, e.g. “the FTIR spectroscopic technique”, not “… method”, as instrumental analytical/physicochemical measurements are termed “techniques”, while a “method” rather implies a set of operations to achieve a purpose); phrases such as “a/the FTIR spectrum” or “FTIR spectroscopic …” are also correct.
Please correct these phrases accordingly in ALL places within the text on the following lines (in the revised version):
Lines 63, 65, 66, 106, 252, 260 (here “FTIR spectroscopic data”), 268 (FTIR spectroscopy), 336, 338, 339 (here it may be replaced by “it” not to duplicate the term), 340 (FTIR spectroscopic facilities)

(Table 1; line 1) In line 1, “=C-H stretching” is irrelevant to the broad asymmetrically shaped composite band with a maximum around ~3300 cm-1. While commonly all aliphatic C–H stretching vibrations are observed as relatively weak bands within a relatively narrow region ~3000–2800 cm-1 (as in lines 2–4 in Table 1; see below), stretching vibrations of C–H moieties in which the carbon atom is involved in a double bond (such as >C=CH2 or >C=CH–) commonly show up at frequencies slightly above 3000 cm-1, e.g. at ~3020…3007 cm-1 (see any reference-type literature on FTIR spectroscopy, e.g. https://doi.org/10.1080/05704928.2016.1230863). In any way, they do not belong to strong broad bands within ~3500–3100 cm-1 which refer to hydrogen-bonded O–H (and in some cases N–H) moieties. So the assignment “=C-H stretching” should be removed from line 1 (Table 1).

(Table 1, lines 2–4) The presented assignments are not correct (e.g., the assignment in line 1 for 2960 cm-1 is not “C-H stretching (symmetric)” but is commonly assigned to antisymm. stretching C–H vibrations in methyl groups (in methylene groups, such vibrations are at lower wavenumbers, ~2930 cm-1); an “aldehyde C-H stretching” band cannot be “symmetric” or “asymmetric”, since in the aldehyde group –CH(=O) there is only one H atom, etc.).  In this region (~3000–2800 cm-1), several separate bands of definite symmetric and antisymmetric C–H vibrations of mainly methyl (–CH3) and methylene (–CH2–) groups (as well as of some methyne groups >CH-) are observed. See, e.g. the reference-type paper (https://doi.org/10.1080/05704928.2016.1230863) or, for particular stretching C–H vibrations, see Table 1 in your Ref. [24] (https://doi.org/10.3390/molecules26041146). Correct the assignments accordingly, taking into consideration that each particular stretching C–H vibration of aliphatic moieties, although appearing commonly within a very narrow wavenumber interval, may still differ by a few wavenumbers in different objects (thus, weak bands at 2960 and 2963 cm-1 may have usually the same assignment, see above).

(Table 1, lines 5 and 6) These bands (1651 and 1541 cm-1) with their characteristic intensity ratio (as in your Fig. 2) for any protein-containing samples are typical amide I and amide II bands, respectively (characteristic of cellular proteins). Please mark them as such (not simply “(amide)”).

(292-294) The following phrase should be corrected (as is specified below): “the marker Fm1465 was situated in the major region of C=O and C=C stretching signals [22] that has been reported as a broad and poorly resolved CH deformation and C-O stretch region…”. While separating your discussion for the two biomarkers, Fm1465 and Fm1729, you evidently did not separate their assignments. Here, the CORRECTED phrase (related to the correct assignment of the band at 1465 cm-1) should be:
“the marker Fm1465 was situated in the major region reported as a broad and poorly resolved C-H bending and C-O stretching region…”.

(301-303) Similarly, the band at 1729 cm-1 has nothing to do with “bending vibrations in methyl/methylene groups”, so the phrase “This wavenumber region can be assigned to bending vibrations in methyl/methylene groups and to stretching vibrations in ester C=O groups, which both are typical for lipids [stuart].” (what is “[stuart]”??) SHOULD BE CORRECTED, e.g. as follows:
“This wavenumber region can be assigned to stretching vibrations of ester C=O groups, which (together with the aforementioned bending C–H vibrations) are typical of lipids.”

(353) “seed/water ratio” – COMMENT: Was it “seed/water mass ratio”?

(353-354) Was the water in the Petri dishes added in the course of the 3-day experiment? (If not, 6 ml of water in non-closed Petri dishes could evaporate at room temperature within a couple of days...) Please specify this in text.

(361-362) “80–90% of field capacity” – What is “field capacity”? Rewrite in a clear way (so that the experiment could be repeated).

MISPRINTS and LINGUISTIC CORRECTIONS:

(118) “such as the drying the leaf tissues” – CORRECT: either “such as the drying of the leaf tissues” (where “the drying” is a noun with the definite article, as the described procedure is meant, with the preposition “of” after it) or: “such as drying the leaf tissues” (where “drying” is a gerund, without an article prior to it and without the preposition “of” after it).

(213) “targets” (correct: “target”).

(291) “biomarkers which were increased their values under” (correct: “biomarkers which increased under”).

(349) Do not capitalise “seeds”.

(574) In Ref. [45], the article number (747) should be given instead of the total number of pages (“1–10” which do not represent any serial page numbers). (Please check other Refs. for such mistakes.)

(SUPPL. INFO – DUPLICATED RECOMMENDATION)

In my previous review report, I gave the following recommendation:

“(SUPPL. INFO) I would recommend to combine all the SI in one PDF file (to facilitate its use).”

 The authors gave the following response to the comment:

“As suggested by Reviewer, supplementary data was combined in one PDF file.”

HOWEVER, as I see from the downloaded Suppl. Info, this has NOT been done, as there are still two PDF files. For the reader it would be more convenient if you combine Supplementary Figure S1 and, below on separate pages, R-scripts for the processing of FT-IR data, in a single PDF file. As is commonly done, please also provide the full paper title (the revised version!) and the list of authors on the first page (above Suppl. Fig. S1).

Author Response

We greatly appreciate your careful consideration and helpful suggestions on our manuscript of the

Manuscript ID: ijms-1598793, entitled “Chemical Fingerprinting of Heat Stress Responses in the Leaves of Common Wheat by Fourier Transform Infrared Spectroscopy”, which was submitted as Article to the International Journal of Molecular Sciences. Please find enclosed the revised version of our manuscript. The portions of the revisions made in the manuscript file are shown in red font.

In the section below, the comments given to the manuscript by the Reviewer are described in italics, and our responses to the comments are described point-by-point.

Comment on the manuscript:

(Lines 63, etc.) The phrase “Fourier-transform infrared (FTIR)” is actually an adjective (with its abbreviation). The whole term you mean (and which should be used in the correct technical language) is “Fourier-transform infrared (FTIR) spectroscopy” (or further, in the abbreviated form, “FTIR spectroscopy” (commonly used without an article) or, e.g. “the FTIR spectroscopic technique”, not “… method”, as instrumental analytical/physicochemical measurements are termed “techniques”, while a “method” rather implies a set of operations to achieve a purpose); phrases such as “a/the FTIR spectrum” or “FTIR spectroscopic …” are also correct.
Please correct these phrases accordingly in ALL places within the text on the following lines (in the revised version):

Lines 63, 65, 66, 106, 252, 260 (here “FTIR spectroscopic data”), 268 (FTIR spectroscopy), 336, 338, 339 (here it may be replaced by “it” not to duplicate the term), 340 (FTIR spectroscopic facilities)

Response to the Comment:

We appreciate very much the Reviewer’s detailed corrections on the usage of the term “FTIR”. According to the suggestions, the phrases were revised to “FTIR spectroscopy” (lines 30, 63, 65, 256, 273, 342, 344 in the 3rd version of the manuscript), “FTIR spectroscopic” (lines 264, 346, 455-456), and “FTIR spectroscopic technique” (lines 67, 70-71, 107, 245). The “It” is used as suggested in the relevant position to avoid the duplication (line 343).

Comment on the manuscript:

(Table 1; line 1) In line 1, “=C-H stretching” is irrelevant to the broad asymmetrically shaped composite band with a maximum around ~3300 cm-1. While commonly all aliphatic C–H stretching vibrations are observed as relatively weak bands within a relatively narrow region ~3000–2800 cm-1 (as in lines 2–4 in Table 1; see below), stretching vibrations of C–H moieties in which the carbon atom is involved in a double bond (such as >C=CH2 or >C=CH–) commonly show up at frequencies slightly above 3000 cm-1, e.g. at ~3020…3007 cm-1 (see any reference-type literature on FTIR spectroscopy, e.g. https://doi.org/10.1080/05704928.2016.1230863). In any way, they do not belong to strong broad bands within ~3500–3100 cm-1 which refer to hydrogen-bonded O–H (and in some cases N–H) moieties. So the assignment “=C-H stretching” should be removed from line 1 (Table 1).

Response to the Comment:

We appreciate the correction by the Reviewer. As suggested, “=C-H stretching” was removed from line 1 in Table 1.

Comment on the manuscript:

(Table 1, lines 2–4) The presented assignments are not correct (e.g., the assignment in line 1 for 2960 cm-1 is not “C-H stretching (symmetric)” but is commonly assigned to antisymm. stretching C–H vibrations in methyl groups (in methylene groups, such vibrations are at lower wavenumbers, ~2930 cm-1); an “aldehyde C-H stretching” band cannot be “symmetric” or “asymmetric”, since in the aldehyde group –CH(=O) there is only one H atom, etc.).  In this region (~3000–2800 cm-1), several separate bands of definite symmetric and antisymmetric C–H vibrations of mainly methyl (–CH3) and methylene (–CH2–) groups (as well as of some methyne groups >CH-) are observed. See, e.g. the reference-type paper (https://doi.org/10.1080/05704928.2016.1230863) or, for particular stretching C–H vibrations, see Table 1 in your Ref. [24] (https://doi.org/10.3390/molecules26041146). Correct the assignments accordingly, taking into consideration that each particular stretching C–H vibration of aliphatic moieties, although appearing commonly within a very narrow wavenumber interval, may still differ by a few wavenumbers in different objects (thus, weak bands at 2960 and 2963 cm-1 may have usually the same assignment, see above).

Response to the Comment:

We thank to the Reviewer for pointing out the mistakes in the FTIR peak annotations. As suggested, the assignments for 2,960, 2,925, and 2,852 cm-1 were revised to “C-H stretching in -CH3 (antisymmetric)”, “C-H stretching in -CH2- (antisymmetric)”, and “C-H stretching in -CH2- (symmetric)”, respectively, in the Table 1. We added two more references in its footnote; one is the ref [24] as suggested, and the other [26] is a new citation of https://doi.org/10.1080/05704928.2016.1230863 that has been suggested above by the Reviewer.

Comment on the manuscript:

(Table 1, lines 5 and 6) These bands (1651 and 1541 cm-1) with their characteristic intensity ratio (as in your Fig. 2) for any protein-containing samples are typical amide I and amide II bands, respectively (characteristic of cellular proteins). Please mark them as such (not simply “(amide)”).

Response to the Comment:

As suggested, the terms “amide I” and ”amide II” were inserted in the lines for 1651 and 1541 cm-1 in Table 1 in the revised manuscript.

Comment on the manuscript:

(292-294) The following phrase should be corrected (as is specified below): “the marker Fm1465 was situated in the major region of C=O and C=C stretching signals [22] that has been reported as a broad and poorly resolved CH deformation and C-O stretch region…”. While separating your discussion for the two biomarkers, Fm1465 and Fm1729, you evidently did not separate their assignments. Here, the CORRECTED phrase (related to the correct assignment of the band at 1465 cm-1) should be:
“the marker Fm1465 was situated in the major region reported as a broad and poorly resolved C-H bending and C-O stretching region…”.

Response to the Comment:

We appreciated very much to the Reviewer for suggesting the corrected phrase. As indicated, the above phrase was inserted in lines 297-298 in the revised manuscript.

Comment on the manuscript:

(301-303) Similarly, the band at 1729 cm-1 has nothing to do with “bending vibrations in methyl/methylene groups”, so the phrase “This wavenumber region can be assigned to bending vibrations in methyl/methylene groups and to stretching vibrations in ester C=O groups, which both are typical for lipids [stuart].” (what is “[stuart]”??) SHOULD BE CORRECTED, e.g. as follows:
“This wavenumber region can be assigned to stretching vibrations of ester C=O groups, which (together with the aforementioned bending C–H vibrations) are typical of lipids.”

Response to the Comment:

We thank the Reviewer again for the corrections. The suggested phrase above was inserted in lines 306-309 in the revised manuscript. We apology our mistake that the [stuart] was for the reference [25]; we forgot to change it to the ref number in the previous manuscript. In the revised manuscript, we added two more references of Kamnev et al. (2021) and Talari et al. (2016) as [24] and [26], respectively, that were used for the annotations.

Comment on the manuscript:

(353) “seed/water ratio” – COMMENT: Was it “seed/water mass ratio”?

Response to the Comment:

Yes, that should be the “mass”. The term “mass” was inserted in line 359 in the revised manuscript.

Comment on the manuscript:

(353-354) Was the water in the Petri dishes added in the course of the 3-day experiment? (If not, 6 ml of water in non-closed Petri dishes could evaporate at room temperature within a couple of days...) Please specify this in text.

Response to the Comment:

The dish was covered by a transparent lid to avoid water evaporation, so that the water was retained during the 3 days. We therefore revised the relevant portion (lines 359-362) as follow:

“The dish was covered by a transparent lid to avoid water evaporation, and the seeds were imbibed for three days at room temperature (25 °C) under a fluorescent room lamp illumination (a light intensity of approximately 10 µmol m-2 s-1) from 9 a.m. to 5 p.m. The germinated seeds were then transferred to pots...”

Comment on the manuscript:

(361-362) “80–90% of field capacity” – What is “field capacity”? Rewrite in a clear way (so that the experiment could be repeated).

Response to the Comment:

We appreciate the Reviewer for pointing out the necessity to clarify the concept of “field capacity”, or FC. The FC is defined as “the amount of water held in the soil after the excess gravitational water has drained away and after the rate of downward movement of water has materially decreased (Assouline and Or, 2014): In the revised manuscript, this article describing the details of concept was cited as a new reference [58] in line 370. To determine the FC-based soil moisture content in the present study, we literally followed a protocol described by Xu and Zhou (2006), therefore we cited this article and inserted a following phrase in line 371 in the revised manuscript:

“The 100% FC was determined as described previously [59].”

Comment on the manuscript:

MISPRINTS and LINGUISTIC CORRECTIONS:

(118) “such as the drying the leaf tissues” – CORRECT: either “such as the drying of the leaf tissues” (where “the drying” is a noun with the definite article, as the described procedure is meant, with the preposition “of” after it) or: “such as drying the leaf tissues” (where “drying” is a gerund, without an article prior to it and without the preposition “of” after it).

Response to the Comment:

Thank you very much for the correction. Among the two options suggested by the Reviewer, we chose the latter “such as drying the leaf tissues” in lines 119-120 in the revised manuscript.

(213) “targets” (correct: “target”).

(291) “biomarkers which were increased their values under” (correct: “biomarkers which increased under”).

(349) Do not capitalise “seeds”.

Response to the Comment:

All the points above were revised accordingly in lines 216, 296, and 355 in the revised manuscript.

(574) In Ref. [45], the article number (747) should be given instead of the total number of pages (“1–10” which do not represent any serial page numbers). (Please check other Refs. for such mistakes.)

Response to the Comment:

In addition to the suggested reference (now ref [46] in the revised manuscript, we found similar mistakes in refs [7], [30], [34], [35], [47], [51]. We corrected these points in the revised manuscript. We thank the Reviewer very much for the corrections.

Comment on the manuscript:

(SUPPL. INFO – DUPLICATED RECOMMENDATION)

In my previous review report, I gave the following recommendation:

“(SUPPL. INFO) I would recommend to combine all the SI in one PDF file (to facilitate its use).”

 The authors gave the following response to the comment:

“As suggested by Reviewer, supplementary data was combined in one PDF file.”

HOWEVER, as I see from the downloaded Suppl. Info, this has NOT been done, as there are still two PDF files. For the reader it would be more convenient if you combine Supplementary Figure S1 and, below on separate pages, R-scripts for the processing of FT-IR data, in a single PDF file. As is commonly done, please also provide the full paper title (the revised version!) and the list of authors on the first page (above Suppl. Fig. S1).

Response to the Comment:

We are sorry for our careless mistake on the SI. This mistake seems to have been caused by the lack of our confirmation of the replacement of the supplementary files in the previous re-submission procedure. In the present re-submission (i.e., 3rd submission), we used a file style format of Int J Mol Sci, and the first page contained the full paper title of the revised version as well as the list of authors above the Suppl. Fig. S1. From the second page, R-scripts were described, so that the whole supplementary information is accommodated in one pdf file, “220301_SI_Osman1.pdf”.

We hope that the revised manuscript is now suitable for publication in the International Journal of Molecular Sciences, and we look forward to hearing from you at your earliest convenience.

Yours sincerely,

Kinya Akashi, Ph.D.,

Professor,

Graduate School of Agriculture Study,

Tottori University.

Tottori 680-8553, Japan.

Tel/Fax: +81-857-31-5352
